# Towards context and domain-aware algorithms for scene analysis

**Ibrahim MOHAMED SEROUIS**                                    *ibrahim.mohamed-serouis@irit.fr*
*CNRS, IRIT*
*Université de Toulouse, Toulouse, France*

**Florence SÈDES**                                    *florence.sedes@irit.fr*
*CNRS, IRIT*
*Université de Toulouse, Toulouse, France*

**Reviewed on OpenReview:** *https: // openreview. net/ forum? id= JQGmbVK4Fr*

## Abstract

Interpersonal interactions and social situations in multimedia content encompass a rich blend of visual, textual, audio and contextual cues as well. However, contextual data integration in multimodal scene analysis research has often been overlooked, leading to incomplete interpretations. For instance, recognizing that two combatants in a video are positioned within a designated ring with a dedicated referee drastically alters the perception from a simple scuffle to a structured martial arts contest.

This paper presents an innovative approach to scene analysis in video content, which not only incorporates contextual data but also emphasizes the most significant features during training. Additionally, we introduce a methodology for integrating domain knowledge into our framework. We evaluate our proposed methodology using two comprehensive datasets, demonstrating promising results compared to a baseline study using one of the datasets. These findings underscore the importance of integrating contextual data into multimodal video analysis, while also recognizing the challenges associated with their utilization.

## 1 Introduction

Within social situations, the human brain engages in a multifaceted analysis of situations and interactions, drawing upon a diverse array of sensory inputs. This cognitive process extends beyond the realm of visual cues, encompassing auditory stimuli, environmental context, cultural influences, and educational backgrounds, thereby contributing to a comprehensive understanding of social dynamics. In the growing landscape of multimodal learning research, a multitude of avenues has emerged, particularly in the realm of human-centric situation understanding and interaction analysis (Qodseya (2020) Kukleva et al. (2020), Liu et al. (2019)).

Nevertheless, the literature has focused mainly on short actions in a sequences of frames (Kondratyuk et al. (2021), Piergiovanni et al. (2022)), and, most recently, as a combination of textual and visual cues (Jiang et al. (2018), Zhang et al. (2022a)) or RGBD data (Xu et al. (2023), Dai et al. (2017)) for studying situations in a video content. However, we consider truthful that a more complete understanding of interpersonal interactions and social situations in a broader sense can only be achieved by incorporating contextual cues such as the context of the scene (for instance, a courtroom drama, a wedding, a rehearsal, or a humorous exchange between characters), its exact location, the relationships between the characters (e.g., friends, lovers, siblings, etc.), or some attributes of the characters such as their occupation, which can influence the power dynamics in the interactions. Also, reasoning on interconnected knowledge for such problems would potentially allow for better interpretability than class activation methods (Selvaraju et al. (2017)), which can be crucial in some scenarios.

**Contributions.** In this research, we provide an interpretable methodology employing interconnected knowledge to capture high- level concepts like scene context and character relationships, while automatically highlighting relevant features at training time. Our methodology is assessed on two scene analysis tasks, yielding promising results. We also provide a framework for integrating domain-knowledge in our methodology, which few of the studies in scene analysis tackle, generalizable to many graph-learning situations.

The goal of our work is to put forward a comprehensive approach to guide systems towards a deeper, higher-level and human-like understanding of interactions between humans. Our findings underscore the relevance of incorporating contextual data in scene analysis and illuminate the challenges that may arise from their usage. To the best of our knowledge, this study one of the rare works attempting to interpret scenes using contextual data, particularly through a knowledge-oriented approach, while also providing methods to integrate cultural factors or domain-specific insights.

**Paper structure:** Section 2 offers an overview of prior research, detailing their contributions, limitations, and our study's positioning. In Section 3, we formally outline the research question and evaluation tasks. Section 4 delves into our approach to processing input data. Our methodology for learning various levels of information is elucidated in Section 5, followed by experimental findings in Section 6. Section 7 elucidates our method for integrating domain knowledge within our framework, followed by a discussion in Section 8 on the limitations of our approach, possible enhancements for our methodology, and the use and extraction of contextual data. Code samples are available at : `https://anonymous.4open.science/r/TMLR-2025-5CE1`

## 2 Related work

**Multimodal scene analysis.** Previous studies in multimodal scene analysis has predominantly focused on the fusion of visual, textual, and audio cues to understand interactions within multimedia content (Stergiou & Poppe (2018), Kukleva et al. (2020), Piergiovanni et al. (2022)) . However, the integration of contextual data, such as the context of the scene or the relationships between the characters involved, has often been neglected in these studies.

**Graph Neural Networks for scene analysis.** Graph neural networks (GNNs) have emerged as a powerful tool for analyzing structured data, including multimedia content (Wu et al. (2020)). They have been successfully applied to tasks such as image classification, object detection, action recognition (Li et al. (2019b), Shi et al. (2019), Zhang et al. (2020), Zhang et al. (2022b)) and video understanding. However, few of these methods included contextual information, and there is a scant exploration of this direction in the literature.

**Context integration in video analysis.** Some recent efforts have begun to explore the importance of contextual information in video analysis. For instance, Wang et al. (2016) proposed a framework for action recognition in videos that incorporates spatial and temporal context. Similarly, Zhang (2023) introduced a method to recognize skeleton-based human activities in videos by considering the scene context as defined by a combination of the action and the location where it occurs, highlighting the potential benefits of integrating contextual data for more accurate analysis of video content. However, these methods did not include additional contextual information such as the relationships between the characters on-screen, or the potential speech during the interactions. They and are limited to action recognition, few provide insights on the importance of each feature, and do not provide a framework on how to include domain-specific knowledge into their methodology.

By building upon and addressing the limitations of existing research, our proposed approach aims to advance the field of multimodal scene analysis by effectively integrating contextual data and possibly domain-specific knowledge insights into the learning process.

## 3 Objective of the framework and evaluation tasks

Consider a graph $G = (V, E)$ consisting of various sets of vertices $V$ and edges $E$ that connect the vertices, such as $E \subseteq \{(x, y) \mid (x, y) \in V^2 \text{ and } x \neq y\}$, where $x$ and $y$ are neighboring nodes. In various applications, such as scene analysis, graphs may exhibit heterogeneous structures, with vertices and edges of diverse types. For instance, nodes representing characters within a scene may convey distinct information

compared to nodes representing the context of the scene. However, in the absence of domain-specific knowledge, discerning the significance of different types of connections or nodes poses a challenge. In the context of a training a deep learning model on such graphs, how can we leverage the varying levels of information encoded within these node sets, and assess the importance of crucial connections or features during training? Moreover, how could we adapt our learning methodology to integrate domain-specific knowledge?

The global objective of our framework is to propose a methodology that integrates various contextual data while addressing this challenge. Our proposed approach will be assessed through two distinct tasks, objectification detection and interaction classification.

## 3.1 Objectification detection

Objectification detection is a high-level interpretative task that consists to predict whether a scene was objectifying or not, meaning that a character was portrayed as a mere object in a scene, either through dialogues that reduce the character to its physical attributes, narratives that treat the character as a prize to be won or a tool to be used, or camera angles that focus on specific body parts. We evaluate our approach on the ObyGaze12 dataset (Tores et al. (2024)), which comprises over 1600 scenes of varying lengths annotated with respect to 4 objectification tags: **Easy Negative (453 samples)** for scenes in which the assessors held a high degree of confidence that objectification was absent, **Hard Negative (711 samples)** for scenes in which the assessors exhibited a reasonable level of confidence regarding the absence of objectification, **Not Sure (397 samples)** for scenes presenting ambiguous characteristics, thereby leading to uncertainty among assessors regarding the presence or absence of objectification, and **Sure (353 samples)** for scenes where the assessors had a strong conviction that objectification was unequivocally present. The scene graphs encompass scenes, places, contexts, characters along with their attributes, and the objectification tags. Domain-specific knowledge and intepretability insights can be crucial for this kind of task, especially juxtaposed to the opinion of expert annotators on the reasons for objectification furnished in the dataset. Classification is performed on the Scene node, which contains the objectification tag.

## 3.2 Interaction classification

Given an input graph containing scenes, places, contexts, characters along with their emotional states, and interactions between characters including their dialogues, this task aims to classify the interactions accurately. We employ the MovieGraphs human-centric situations dataset (Vicol et al. (2018)) for evaluation, which offers graphs with 101 annotated interaction classes (*e.g asks, fights (with), hugs, threatens, explains (to)…*) spanning more than 17,000 instances. It is a wide collection comprising 7637 scenes of varying lengths from 51 movies (with release dates ranging from 1945 to 2014) represented in a graph format. These graphs encapsulate comprehensive scene-specific information, encompassing character identities, physical and personal/filmic attributes (such as sex, occupation, age, ethnicity, appearance), character relationships, interpersonal interactions, and chronological markers. Additionally, the dataset includes separated subtitle files for each movie. For exhaustive details on the dataset, refer to the original study. Classification is performed on the Interaction node, which contains the interaction class.

# 4 Dataset processing

## 4.1 Speech features

Although the identity of the speaker is not mentioned in both datasets, we extract the dialogs and map each interaction with the corresponding dialogs if there is an overlap lasting more than one second or equivalent to the duration of the interaction. Subsequently, we generate the representation of dialogs using a BERT (Devlin et al. (2019)) embedding model.

More specifically, we run an inference on each line for each dialog on the tokenized version of the speech file. Subsequently, leveraging a Transformers (Wolf et al. (2020)) BERT embedding model for sequence classification, we extract the hidden states of all the lines. Finally, we perform a Max Pooling operation (Gholamalinezhad & Khosravi (2020)) over one dimension of the text embedding here obtained, and pass the

output to two fully-connected (dense) layers with relu and tanh activations, to obtain the final representation. While more recent models exist for extracting textual features, models using the simpler BERT-base uncased produced better results for both tasks.

### 4.2 Interaction graphs generation

MovieGraphs dataset (Vicol et al. (2018)) contains scene graphs with at least one interaction per graph. Meanwhile, one of our research objectives entails the classification of individual interactions.

To address this challenge, we have devised an approach that involves the creation of a dedicated graph for each interaction, guided by two fundamental rules: (1) characters who are not involved in the current interaction are not included in the interaction graph, and (2) any element present in the original scene graphs that pertains to the entirety of the scene or lack a chronological indicator (timestamps), will be linked to each corresponding derived interaction.

Subsequently, the interactions and relationship classes are grouped respectively in 101 and 15 similar categories, identically from our baseline study (Kukleva et al. (2020)). This process results in an interaction graph as shown in Fig 1, which is generated from the complete scene graph and retains exclusively pertinent or interconnected information. The here obtained graphs are then serialized and stored in a *.tfrecord* files, which will serve as the data source for training the Tensorflow (Abadi et al. (2015)) models.

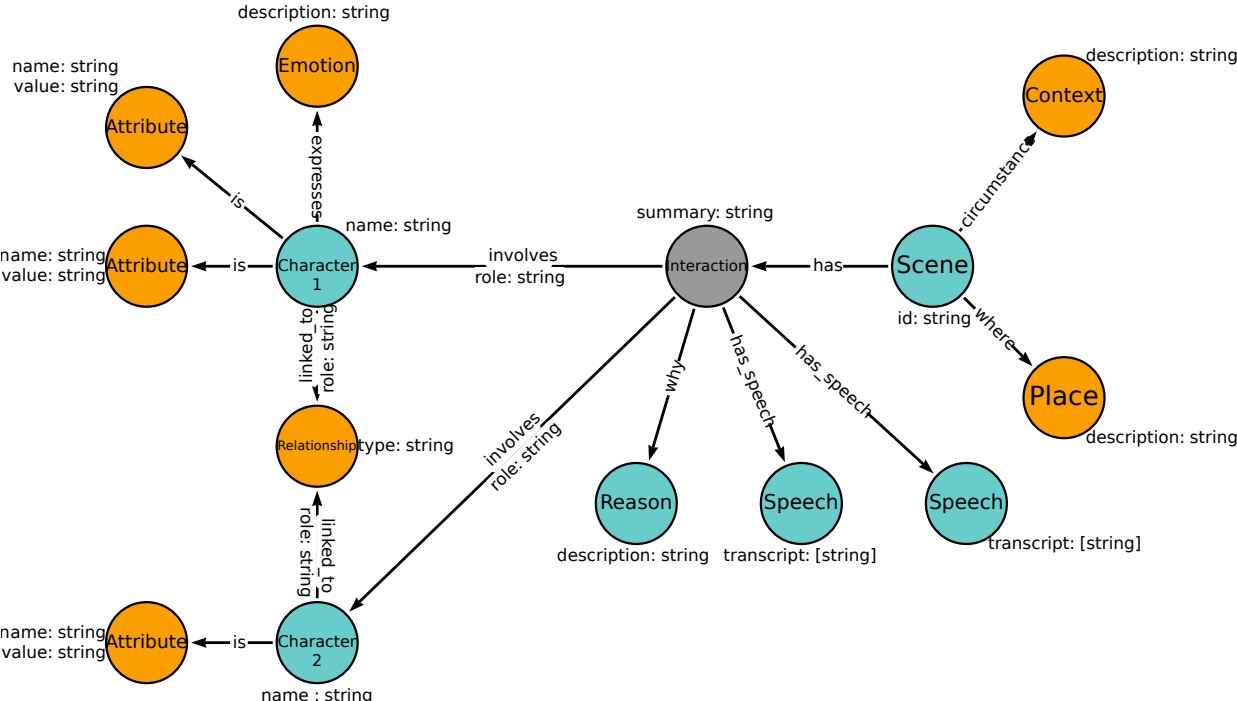

Figure 1: **Typical interaction graph.** The grey node represents the root node (Interaction Entity). All the nodes represent elements that pertain to the entire scene or, at least, the singular interaction.

### 4.3 Data split

For the interaction classification task, we reproduced the same data split as in the baseline study, LiREC (Kukleva et al. (2020)), leading to roughly 12000 training samples and 4000 validation samples. For the objectification detection task, 60% percent of the dataset was allocated for model training, while the remaining 40% was reserved for the validation of results. It is worth noting that alternative partitioning strategies could have been employed had a larger dataset been available.

# 5 Proposed framework: learning process

In this section, we introduce our methodology, a three-step approach grounded in Graph Neural Networks (GNNs). It offers a versatile pipeline for training on various levels of information and exploring various graph architectures, without substantial deviations from the foundational structure. It establishes a cohesive framework for the models, founded on the principles of feature mapping, as expounded in 5.1, followed by graph updates, elaborated in 5.2, and culminating in the classification and interpretability processes, detailed in 5.3 and 5.4.

The model's architecture is shown in Figure 2, illustrating the approach for objectification detection. The sole distinction between the two methodologies lies in the penultimate dense layer: it features a kernel size of 512 for objectification detection and 1024 for interaction classification. These parameter values were selected based on achieving optimal experimental results.

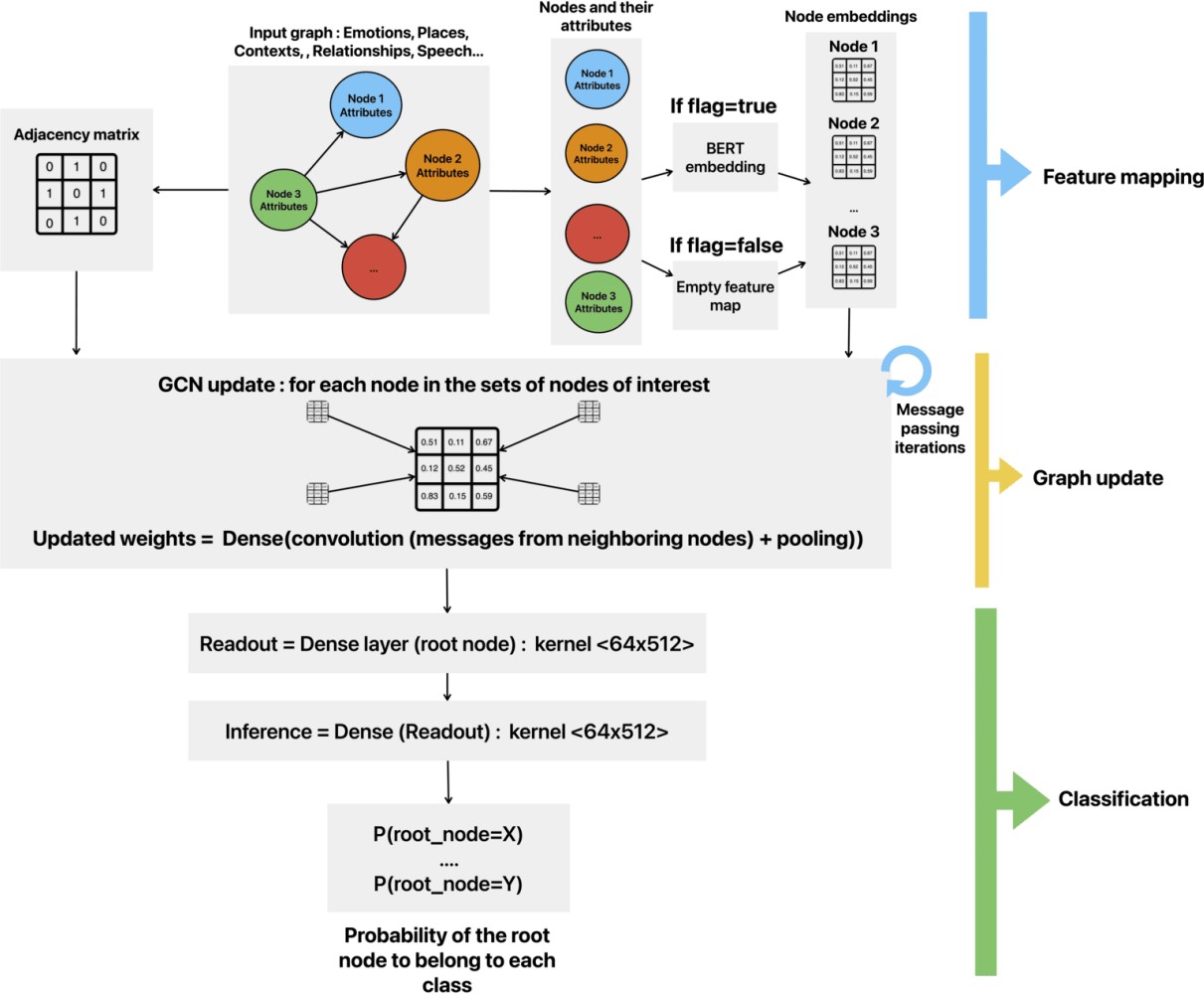

Figure 2: **Proposed framework.** Each phase of the proposed framework is depicted by at least one layer (rectangular boxes), and its shape is explicitly defined within the '<>' symbols.

## 5.1 Feature mapping

The feature mapping phase is what separates our framework from other graph neural networks learning cycles (i.e Feature Mapping, then Graph Updates, then Classification).

In this preliminary stage, we establish a set of transformations designed for the individual nodes or edges within our graph. These transformations are contingent upon the particular type of node or edge, enabling us to craft the feature representation through tailored adjustments.

**The selection of node and edge types:** the models have been trained on the $2^7$ potential combinations of input node states (with 7 corresponding to the types of nodes in both datasets), utilizing an automated script and training flags to specify the usage of particular node information during the training process. We loop through the list of nodes sets, check the value of the flag for each node type, then assign its representation based on the flag. The training flags exhibit binary values, with *false* denoting absence of information pertaining to the current node (we then generate an empty or masked tensor), denoting the presence of an empty information that shouldn't be treated as proper weights, or *true*, meaning that a proper representation will be computed using a $<50000\text{x}32>$ embedding layer. This process is detailed in Algorithm 1 and Algorithm 2.

**The initialization of the initial state of each node type:** Node sets other than the subtitles are embedded through an embedding model with a vocabulary size of 50000 and a dense embedding size of 32, yielding their initial state.

This phase culminates in the generation of the graph representation matrix which is derived from the combination of the adjacency matrix and the nodes features set vector.

## 5.2 Graph update

Herein, we make decisive architectural choices that tailor the model to our algorithm. The four primary architectural designs for Graph Neural Networks were explored for our specific problem: and Graph Attention Networks (Dwivedi & Bresson (2021)), Graph Convolutional Networks (GCN) (Kipf & Welling (2017)), GraphSAGE networks (Hamilton et al. (2017)) and Vanilla Message Passing Neural Networks (Vanilla MPNN) (Gilmer et al. (2017)), with layer support provided by Tensorflow-GNN (Ferludin et al. (2022)). Further insights into each architecture are elaborated in Appendix A.

The Graph Update phase iteratively refines the node and edge features, enabling the GNN to learn more abstract and informative representations from the graph structure and the feature data. During our graph update phase, masked or empty nodes are updated based on the aggregated information (embeddings) from their neighboring non-masked nodes.

This phase ultimately yields a collection of hidden states corresponding to each pair of nodes and edges within the graph representation. This collection is represented by a layer whose configuration varies in accordance with the chosen model architecture. The number of the parameters on this phase vary depending on the specified training flags.

## 5.3 Classification

Final step, based on the previously acquired representation, it consists in the determination of the class assigned to the root node (*Interaction* node for interaction classification, *Scene* node, for objectification detection).

Comprising two sub-steps, this phase maintains a uniform structure, irrespective of the selected *graph update* technique:

1. Retrieving the *hidden state* of the *root node* produced through the graph update phase, with a readout function.

2. Strictly speaking, classification of this state via a fully connected layer or classification head: prediction of a class of the root node. Its formal definition is as follows:

$$\hat{y} = R(H_v^T | \{v\} \in G) \tag{1}$$

Where $\hat{y}$ is the prediction, $\boldsymbol{R}$ the readout function, $\boldsymbol{H_v^T}$ the hidden states for the node sets in $\mathbf{v}$, and $\mathbf{v}$ the sets of nodes used for the readout.

This phase yields the probability of the graph belonging to each class as its output.

### 5.4 Interpretability insights

In addressing the inherent black-box nature of Deep Learning models, known for producing results without clear interpretability (Hussain (2019)), we introduce an interpretability sub-module. This sub-module aims to elucidate the influence of input features on the final output of the model.

To derive these insights, we calculate the sensitivity or gradients of the last hidden states, with respect to individual elements of the weight matrices in the previous layers. It provides a global view of the impact of each input element on the output. This operation can be articulated as follows:

$$Sensitivity = \frac{\partial h^{(l+1)}}{\partial W_{pq}^{(l)}} \tag{2}$$

Where ($h^{(l+1)}$ are the hidden states of the layer $l+1$ (last layer), $\mathbf{p}$ and $\mathbf{q}$ the indexes for respectively the rows and the columns of the weight matrix, and $W_{pq}^{(l)}$ the element at the p-th row and the q-th column of the weight matrix $W^l$.

This computation results in a sensitivity matrix for each weight matrix, where the elements are aggregated by the mean of the elements, akin to some implementations of GRAD-CAM (Selvaraju et al. (2017)), yielding a score. Given that the messages during the graph update phase may possess diverse dimensions, each sensitivity score is normalized by the reciprocal of the number of rows in its corresponding message:

$$Score = Sensitivity \cdot \frac{1}{rows} \tag{3}$$

## 6 Experimentation and results

### 6.1 Implementation details

**GNN implementation.** In all considered architectures, key parameters remain consistent, including the dimensionality of message exchanges between nodes and their neighbors across graph update layers, set at 64. Similarly, the number of iterations for message passing remains uniform at 1, as does the message pooling method, employing summation across neighboring nodes. For GraphSAGE, GCN and GAT models, the common choices are related to the nodes where the convolution (for GCN and GAT) or neighborhood sampling and aggregation (for GraphSAGE) are computed. For our study, we convoluted or sampled over the 3 nodes of interest: Interaction, Character, and Scene, chosen due to their extensive connectivity with other graph components.

**Loss function.** Our models are trained using sparse categorical cross entropy as loss function (Mao et al. (2023)), which is a variation of the (categorical) cross entropy that uses integers instead of one-hot vectors for the classes outputs. Its formal definition is as follows:

$$J(w) = -\frac{1}{N} \cdot \sum_{i=1}^{N} [y_i \log(\hat{y}_i) + (1 - y_i) \log(1 - \hat{y}_i)] \tag{4}$$

Where $N$ is the number of samples, $y_i$ represents the true class, and $\hat{y}_i$ the predicted class.

**Training parameters.** The batch size for interaction classification was adapted to our baseline study, LIREC (Kukleva et al. (2020)). Both models were trained using an Adam optimizer for 200 epochs; however, training was automatically halted using early stopping (Yao et al. (2007)) once the monitored value (accuracy or binary accuracy) on the validation set exhibited no further improvement for a consecutive period of 40 epochs. For a comprehensive overview of the model implementations, please refer to Table 1, which summarizes the pertinent details.

Table 1: Implementation details for both tasks

(a) Interaction classification

| Parameter | Value |
|---|---|
| Optimizer | Adam |
| Learning rate | $3*10e^{-5}$ |
| Batch size | 64 |
| Early stopping | top-5 accuracy |
| Epochs | 200 |
| Dropout rate | 0.1 |

(b) Objectification detection

| Parameter | Value |
|---|---|
| Optimizer | Adam |
| Learning rate | $cosine\ decay(10^{-4},\ 4800\ steps)$ |
| Batch size | 32 |
| Early stopping | Validation accuracy |
| Epochs | 200 |
| Dropout rate | 0.2 |

## 6.2 Evaluation metrics

**Accuracy.** Used for both tasks, it quantifies the frequency at which the model achieves a correct prediction, meaning that the class predicted by the model corresponds to the actual class. The mathematical expression for computing this metric is detailed as follows.

**Top-5 accuracy**. Variation of the top-n accuracy metric, this approach assesses how often the class predicted by the model falls within the top 5 probabilities when a prediction is indeed accurate. We use it to match the metrics given by our baseline study. In the context of our problem where classes are loosely annotated, such as *asks, answers, talks*, and *explains*, the relevance of top-5 accuracy becomes evident. This metric proves valuable because distinguishing between the latter two classes can be particularly challenging even for a human.

**Loss (cross-entropy loss), as defined in Section 6.1.** Used for objectification detection, it measures the difference between two probability distributions: the true distribution and the predicted distribution.

**Area Under the Curve (AUC).** We use it to evaluate the objectification detection task. When presented with a randomly chosen positive and a randomly negative instance, the Area Under the Curve (AUC) represents the probability that the classifier possesses the capability to discern their respective classes (Fawcett (2006)). It is the area under the Receiver Operating Characteristic (ROC) curve, which is a graphical representation that illustrates the performance of the True Positive Rate (TPR) in comparison to the False Positive Rate (FPR) (Monaghan et al. (2021)) at various threshold values $T$. The AUC can be formally defined as:

$$AUC = \int_{x=0}^{1} TPR(FPR^{-1}(x))dx \qquad (5)$$

When determining the best models, a comprehensive evaluation considers all of these metrics. For instance, models exhibiting elevated accuracy rates in both training and validation stages, yet possessing an AUC score $\leq 0.5$ are deemed ineffective, due to their inability to provide discrimination capacity beyond that of random chance.

**F1 score.** It is a measure of a test's accuracy that combines both precision and recall. It ranges from 0 to 1, with a higher score indicating a more accurate test. It's calculated as:

$$F1 = 2 \cdot \frac{(Precision * Recall)}{(Precision + Recall)} \qquad (6)$$

## 6.3 Results

### 6.3.1 Objectification detection

Based on the results presented in Table 2a, when considering all pertinent metrics, our best models have demonstrated the capability to attain accuracy levels substantially exceeding those of random chance (*i.e*, a 50% accuracy score), indicative of their efficacy on the binary classification task. However, due to the

class imbalance with 353 instances categorized as *Sure* and 453 as *Easy Negative*, we cannot solely rely on accuracy. Therefore, the utilization of the AUC metric is warranted in addressing this issue. Leveraging the ROC curve, as advocated in previous empirical investigations (Fawcett (2006)) for its capacity to greatly but not completely decouple classifier performance from class skew, they also exhibit a great level of performance, exceeding those of a random classifier (*i.e*, a 0.5 AUC score). It is noteworthy that GCN (graph convolutional networks) models, using a combination of contextual (*i.e, scene graphs)* data and speech data, exhibited superior intrinsic performance compared to our baseline Tores et al. (2024), and MPNNs showed F1-score levels similar to our baseline.

In order to mitigate class imbalance, we oversampled the minority class (easy negative examples). We managed to obtain better results for all the models, except for GraphSAGE models. As we can observe in Table 2b, GCN and GraphSAGE models did not need the introduction of speech features to achieve better results when using oversampling. However, all the models shown a slight rise in their validation loss, but they managed to outperformed our baseline study in terms of F1-score.

Table 2: **Best results on the validation set for the objectification task, for each architecture. (SG=Scene Graphs, S=Speech, I=Image, NC=Not Communicated)**

(a) Results without oversampling

| Model | Data | Accuracy (%) | AUC | F1-score | Loss |
|---|---|---|---|---|---|
| GCN | SG + S | **65.62** | **0.64** | **0.81** | **0.70** |
| Tores et. al | I | N.C | N.C | 0.79 | N.C |
| MPNN | SG + S | 60.93 | 0.63 | 0.79 | 0.76 |
| SAGE | SG | 63.28 | 0.62 | 0.77 | 0.81 |

(b) Results with oversampling

| Model | Data | Accuracy (%) | AUC | F1-Score | Loss |
|---|---|---|---|---|---|
| GCN | SG | **66.40** | **0.69** | **0.81** | **0.76** |
| MMPNN | SG + S | 62.10 | 0.66 | 0.8 | 0.90 |
| SAGE | SG | 61.32 | 0.6258 | 0.8 | 0.75 |
| Tores et. al | I | N.C | N.C | 0.79 | N.C |

Utilizing the sensitivity scores delineated in Section 5.4, we can offer insights into the decision-making process of our models. Based on the scores provided in Table 3a, we can infer that our optimal model (GCN) assigns greater significance to the weights associated with the activity or context of the scene and the attributes of the featured characters, while attributing minimal importance to biases related to the location of the scene in the decision-making process. It is important to note that the model's heightened emphasis on the context is promising, aligning with its prominence in the decision-making process observed by expert annotators (Tores et al. (2024)), as outlined in Table 3b.

(a) **Sensitivity scores for the best model**. (W=Weights, B=Biases.)

| Unit | Score |
|---|---|
| (W) Activity/Context of the scene | 3.27 |
| (W) Attributes (Body, Clothes, Occupation) | 2.12 |
| (B) Location of the scene | 0.015 |
| (W) Relationship between characters | 0.004 |

(b) Top 4 reasons (concepts) for objectification highlighted by expert annotators

| Reason | Appearances |
|---|---|
| Speech | 410 |
| Activities | 156 |
| Clothes | 112 |
| Body | 86 |

Table 3: Comparative analysis of sensitivity scores and top reasons for objectification

### 6.3.2 Interaction classification

The findings presented in Table 4 demonstrate that, by exclusively leveraging contextual data, we can achieve top-1 and top-5 accuracy levels that outshine the performance achieved by utilizing both visual and dialog data. This observation underscores the significance of contextual information in our classification task. Moreover, the incorporation of dialog data resulted in notable enhancements in training and validation accuracy. The rise in raw accuracy can be attributed to the enhanced performance achieved for certain prevalent verbal interactions within the dataset. An illustrative example is the *asks* class, which accounts for approximately 14% of the training and validation set.

Table 4: **Results on the validation set for the best models, sorted by top-1 accuracy (V=Visual, D=Dialogs, F=Face tracks, SG=Scene Graphs). To ensure a fair comparison, we adopted the same batch size, training, validation, and test data splits as our baseline study, LIReC (Kukleva et al. (2020)).**

| Model | Data | Accuracy (%) | Top-5 accuracy (%) |
|---|---|---|---|
| LIReC (baseline) | V | 18.7 | 45.8 |
| LIReC | D | 22.4 | 50.6 |
| LIReC | V + D | 25 | 54.8 |
| LIReC | V + D + F | 26.1 | 57.3 |
| GCN | SG + S | **33.4** | **68.12** |

**Ablation study**

Our ablation study focuses on objectification detection. In this section, we will compare the results on the validation set for the models without each contextual cue, and juxtapose them to the results of the sensitivity scores outlined in Table 3.

The results presented in Table 5 are largely consistent with our sensitivity scores calculations. Models without the strongest cue, the context of the scene, yielded the poorest overall performance (marginally lowest AUC and accuracy). The second-worst results were observed when the characters' attributes were eliminated. A slight improvement in results was achieved by removing only the scene location, while the best results were obtained by excluding the scene location, and the relationship between the characters. These findings could potentially corroborate our insights derived from the sensitivity scores for each contextual cue.

Table 5: Mean results on the validation set for the ablation study.

| Removed cue | Accuracy (%) | AUC | Loss |
|---|---|---|---|
| Activity/Context of the scene | 55.98 | 0.50 | 0.74 |
| Attributes (Body, Clothes, Occupation) | 56.09 | 0.54 | 0.76 |
| Location of the scene | 56.39 | 0.54 | 0.75 |
| Relationship between the characters | 57.22 | 0.54 | 0.75 |

## 7 Integrating domain-knowledge

In various real-world scenarios such as healthcare diagnostics or financial risk assessment, domain knowledge plays a pivotal role. Similarly, in our objectification detection task, understanding the nuances of objectification and its contextual implications is crucial. However, integrating domain knowledge into complex architectures poses a significant challenge, and few studies have addressed this aspect in their proposed methodologies. Aside from traditionally assigning edge weights, few studies approach this problem for Graph Neural Networks. To address this gap, we propose two practical approaches for seamlessly integrating do-

main knowledge into our framework: (1) Node Ranking, detailed in Section 7.1, and (2) The probabilistic approach, explained in Section 7.2.

## 7.1 Node ranking

Collaborating with domain experts enables us to discern the significance of specific nodes within the dataset. By assigning ranks to nodes based on the importance of the information they represent, we can then impose constraints on the weights of certain layers. For instance, we may stipulate that the mean absolute value of weights associated with higher-ranked nodes ($m_h$) should exceed those linked to lower-ranked nodes ($m_l$). If the condition is not satisfied, we penalize the weights $\boldsymbol{W_i}$ of lower-ranked nodes by $\boldsymbol{\alpha}$ values, such as:

$$W_i = W_i \cdot \frac{m_h}{m_l} \tag{7}$$

Consequently, the contributions of highly ranked nodes to the overall inference process would be prioritized over those of less ranked nodes. These constraints can be seamlessly integrated into popular graph neural network (GNN) frameworks found in the literature, such as DGL (Wang et al. (2019)), Torch-Geometric (Fey & Lenssen (2019)), or Tensorflow-GNN (Ferludin et al. (2022)). Howver, early iterations of this process did not manage to increase the results, although performed without domain experst.

## 7.2 Probabilistic neuro-symbolic approach

Given some specific conditions, we can create a probabilistic problem in the form of a Bayesian Network exploiting the outputs of the models, with tools such as Problog (De Raedt et al. (2007)).

For instance, in the context of objectification detection, the model output (probability of the scene being objectifying) can serve as the initial probabilistic fact (prior knowledge). Subsequently, we can update this knowledge by incorporating domain-knowledge insights regarding the likelihood of a scene being objectifying given certain factors, such as the presence of a naked woman or the depiction of a domestic activity.

The core concept of this idea involves establishing a set of probabilistic facts $\boldsymbol{P_f}$ to refine prior knowledge (in our case, specifically the likelihood of a scene being objectifying). For each probabilistic fact $\boldsymbol{Pf_i}$, we define a confidence interval $[\boldsymbol{Pf_i - Pf_i * \alpha} \; ; \; \boldsymbol{Pf_i + Pf_i * \alpha}]$. The process then involves performing inferences on the values within this interval, beginning at the lower bound (or start and the higher bound) and incrementally increasing/decreasing each probabilistic fact $\boldsymbol{Pf_i}$ by a $\boldsymbol{k}$. The process can be halted by early stopping if the results do not increase according to the selected metrics.

An examination of the distributions of the results pre and post-update using KL divergence (Kullback & Leibler (1951)) for instance would provide insights on the updated results. We can also infer uncertainty on the updated distribution post-application using uncertainty estimation (Shannon (1948)).

Early iterations of this process, applied to the objectification detection task, managed to increase both accuracy (from 66.2 to 70.2) and AUC score (from 0.69 to 0.81) metrics. However, it was done without a team specialized in the matter, which can be crucial for the task. Nonetheless, we are currently collaborating with experts in social sciences and humanities to integrate more expert domain-specific knowledge. The architecture of the proposed framework is available in Appendix E .

# 8 Discussion

## 8.1 Limitations

For objectification detection, top-performing models (GCNs) consistently struggled in scenes involving specific combinations such as [Type of Plan+Soundtrack], [Soundtrack+Voice], or solely Soundtrack or Voice as reasons for objectification (less than 10% accuracy), possibly due to insufficient data. Also, certain information such as the type of plan and narratology cannot be effectively used by the models.
For interaction classification, top-performing models consistently failed on detecting interactions heavily influenced by visual factors, such as *walks*, *grabs*, or *holds.*

Also, our interpretability framework at the moment fails to capture the importance of speech nodes for Graph Convolutional Networks. This might be due to some instances with unconnected gradients between the embedding of the speech and the final output, or NaN losses in a convolution layer. We aim to mitigate this effect on future iterations.

## 8.2 Integrating image features

Initially, we attempted integrating frame features extracted using X-Clip (Ni et al. (2022)) for computing the representation of the clip directly into model training, by introducing a "frame" node. However, this method resulted in a significant increase in loss for the objectification detection task, from 0.8 to an average of 10, with only a marginal 0.1% improvement in accuracy. Moreover, it substantially elevated memory consumption during graph embedding and update phases, nearing the limits of our computing unit.

For the second approach, we trained the models on frame features for each movie and combined outputs with a dense layer. While this approach led to reduced errors, it resulted in a slight decline in binary accuracy. Additionally, we encountered challenges post-training, particularly in computing gradients for models, and sometimes, training failures due to kernel crashes caused by a memory leak. Similar trends were also observed in the interaction classification task.

The absence of improvement with the incorporation of image features may be attributed to the elements of the graphs being annotated predominantly by annotators after thorough video analysis, as stated in the original study by Vicol et al. (2018), leading to potential redundancy in information.

## 8.3 Contextual data extraction for scene analysis

Acquiring contextual data, particularly regarding character relationships, is more feasible within film data compared to typical social scenarios. However, the associated costs may hinder extensive research in this direction. For instance, the MovieGraphs dataset offers comprehensive human-annotated classifications, curated through labor-intensive processes. Yet, this approach may not be accessible to all researchers due to its time and resource intensity. Alternative methods, such as employing visual place recognition algorithms (Zhang et al. (2021)) or leveraging contemporary computer vision techniques for age and gender extraction (Huang et al. (2017); Dong et al. (2016); Jia et al. (2016)), could streamline and automate data collection. Additionally, emotion and relationship recognition algorithms (Ko (2018); Kukleva et al. (2020)) provide further insights. However, the context of the scene remains hard to extract in the cases where metadata are not provided or human input is not present.

While AI models may accelerate data annotation, human intervention remains crucial for specifying character relationships. AI may struggle with nuanced contextual interpretations, such as distinguishing between genuine and faked emotions. Thus, human input is indispensable for refining extraction outputs. Also, it is important to pay attention to the potential for cumulative errors with multiple extraction algorithms. While contextual data can yield a deeper level of interpretation, it is imperative to emphasize the crucial role played by the establishment of precise and unambiguous guidelines for annotating and collecting data, in their successful integration into the realm of multimodal scene analysis and deep learning. Notwithstanding, a comprehensive pipeline for extracting contextual data will be addressed in a forthcoming publication.

## 9 Conclusion

This research has been dedicated to the integration of contextual cues for scene analysis, driven by the imperative to augment the comprehension of social scenarios by computational systems beyond the traditional use of image and text. To address this, we introduced an innovative approach that integrates contextual information into our analysis pipeline, yielding better results than a state-of-the-art algorithm on an interaction classification task.

Starting from the promising results here obtained, significant improvements would be achievable with more data-centric approaches. In particular, we could reduce the human-annotated target classes within the

interaction classes to enhance the generalization capabilities of the model, or iterate on a more balanced dataset.

Moreover, an intriguing avenue for future investigation involves proposing a pipeline to automate the extraction of contextual data using cutting-edge techniques. Additionally, we aspire to collaborate with experts in objectification and various social science domains to leverage their domain-specific insights in refining our methodology. We aim to explore these aspects in our future research endeavors, and apply our framework to diverse datasets and human-centric situations tasks.

**Broader Impact Statement**

As our methodology continues to evolve, we anticipate expanding its scope to tackle pertinent societal issues such as the depiction of sexism and racism in films or scenes. By employing artificial intelligence techniques for high-level interpretative tasks in scene analysis, we aim to contribute to the advancement of a media landscape characterized by inclusivity and respect for all genders and races. This integration into production, evaluation, and moderation practices holds promise for reshaping the media environment positively.

Failure to address these aspects meticulously may result in perpetuating biases present in the training data, leading to misinterpretations or the reinforcement of existing stereotypes. This, in turn, could inadvertently exacerbate social issues rather than ameliorating them. Possible solutions that we may envisage for mitigating this issue are:

- **Diverse and representative data**: ensuring that training datasets are diverse and representative of various demographics, cultures, and perspectives. This helps mitigate bias by providing a more comprehensive understanding of the data distribution.

- **Regular audits and evaluations**: conducting regular audits and evaluations of AI systems to assess their performance, identify any biases or unintended consequences, and take corrective actions as necessary.

- **Education and awareness**: providing education and training programs on bias, diversity, and inclusion for AI developers, users, and stakeholders. This helps raise awareness of potential biases and encourages proactive efforts to address them.

- **Transparency and accountability**: maintaining transparency in the development and deployment of AI systems, including disclosing the data sources, algorithms used, and decision-making processes. Establish mechanisms for accountability to address any unintended consequences or biases that may arise.

- **Privacy protection**: As we deal with human content, data should be anonymized in order to safeguard individual's entities and ensure ethical data handling practices throughout the research process.

**Acknowledgments**

This work has been supported by the French National Research Agency through the ANR TRACTIVE project ANR-21-CE38- 00012-01.

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

# A    Graph Neural Networks architectures explored

As stated in Section  5.2, four primary architectural designs for Graph Neural Networks were explored for our specific problem: and Graph Attention Networks (Dwivedi & Bresson (2021)), Graph Convolutional Networks (GCN) (Kipf & Welling (2017)), GraphSAGE networks (Hamilton et al. (2017)) and Vanilla Message Passing Neural Networks (Vanilla MPNN) (Gilmer et al. (2017)). In the subsequent paragraphs, we will elaborate on the principles of each architecture, and provide few additional implementation details.

**GCN models.** These architectures apply the principles of convolution (LeCun et al. (1998)), as observed in conventional Neural Networks, to graphs. They enable learning representations of nodes in a graph that incorporate information from their local neighborhood and can capture complex relationships and structures within the graph. There are many variations of the original GCN architectures in the literature, such as DeepGCN (Li et al. (2019a)) or Fast-localized spectral GCN (Defferrard et al. (2016)), but our graph convolutional layer corresponds to (Kipf & Welling (2017)) that computes:

$$\mathbf{x}_i^{(k)} = \sum_{j \in \mathcal{N}(i) \cup \{i\}} \frac{1}{\sqrt{\deg(i)} \cdot \sqrt{\deg(j)}} \cdot \left( \mathbf{W}^\top \cdot \mathbf{x}_j^{(k-1)} \right) + \mathbf{b} \tag{8}$$

Where $\mathbf{W}$ is weight matrix applied to the node features, $x_j^{(k-1)}$ the node features of node j in layer **(k-1)**, **b** the bias vector applied to the aggregated output, and **i** and **j** the neighboring nodes.

**Vanilla MPNN models.** They exchange information between nodes through message passing steps, updating node representations based on information from neighbors, without convolution. The Vanilla MPNN update layer can be formally defined as (Gilmer et al. (2017)):

$$h_v^{k+1} = U_k(h_v^k, m_v^{k+1}) \tag{9}$$

Where $h_v^{k+1}$ are the hidden states at layer $k+1$, $U_k$ the node update function, $h_v^k$ the hidden states at layer $k$, and $m_v^{k+1}$ the message at layer $k+1$.

**GraphSAGE models.** They work by sampling and aggregating information from the neighborhood of each node iteratively. In each iteration, GraphSAGE samples a fixed number of neighbors for each node, aggregates their features, and updates the node's representation. The GraphSAGE update layer can be formally defined as (Hamilton et al. (2017)):

$$h_v^k = \sigma(W^k.\mathbf{concat}(h_v^{k-1}, h_{N(v)}^k)) \tag{10}$$

Where $\sigma$ represents the nonlinear activation function, $W^k$ the weight matrix at layer k, **concat** the function that concatenates the representation of the node and its aggregated representation, $h_v^{k-1}$ the representation of the node at layer k-1, and $h_{N(v)}^k$ the aggregated representation of the node and its immediate neighborhood.

**GAT models.** These architectures apply the principles of Attention (Vaswani et al. (2017)) in traditional neural networks to graphs. They use attention mechanisms to weigh the importance of neighboring nodes' features when updating a node's representation. The GAT update layer can be formally defined as (Dwivedi & Bresson (2021)):

$$h_i^{(l+1)} = \sigma \left( \sum_{j \in \mathcal{N}(i)} \alpha_{ij} \cdot W^{(l)} \cdot h_j^{(l)} \right) \tag{11}$$

Where $h_i^{(l+1)}$ is the updated representation of node $i$ at layer $l+1$, $\sigma$ is the activation function, **ReLU** (Nair & Hinton (2010)) in our case, $\mathcal{N}(i)$ represents the neighbors of node $i$, $\alpha_{ij}$ represents the attention weight between nodes $i$ and $j$, $W^{(l)}$ is the weight matrix for layer $l$, and $h_j^{(l)}$ is the representation of node $j$ at layer $l$.

# B  Algorithm: generation of the training flags

---

**Algorithm 1** Training flags generation

---

**Require:** $node\_sets \neq null$, $size(node\_sets) = 7$
  ▷ **0. Set the possible values for the training flags**
  $values \leftarrow [True, False]$
  ▷ **1. Generate the combinations (with repetition) of the values for a size = size(node_sets)**
  $train\_flags \leftarrow cartesian\_product(values, repeat = size(nodes\_list))$

  ▷ **2. Loop through the training flags and compute the representation**
  **for** $flag\ in\ training\_flags$ **do**
      $repr \leftarrow$ **feature_mapping**$(node\_sets,\ flag)$
  **end for**
  ▷ **Description:** The training flags exhibit binary values, with *false* denoting absence of information pertaining to the current node, or *true*, meaning that a proper representation will be computed using the <50000x32> embedding layer. Given the existence of seven distinct node types in our representation, we generate combinations of lists with a size of seven, comprising elements "true" and "false." Subsequently, we iterate over the training flags list to compute the representation for each node, contingent upon the flag value corresponding to its respective type.

---

# C  Algorithm: Computing the representation of the nodes

---

**Algorithm 2** Computing the representations of nodes

---

**Require:** $node\_sets \neq null$, $flag \neq null$
  ▷ **Loop through the list of nodes sets and check the value of the flags for each node type and assign its representation based on the flag**
  **for** $node\ in\ node\_sets$ **do**
  ▷ **Check the value of the flags for each node type and assign its representation based on the flag**

      **if** $flag[node['type']] = True$ **then**
          $representation[node] \leftarrow$ **embedding**$(node)$
      **else**
          $representation[node] \leftarrow$ **empty_tensor**$()$
      **end if**
  **end for**
  $return\ (representation)$

  ▷ **Description:** While looping through the list of nodes, based on the value of the training flag for the current node, we compute an empty feature map or send the information through an embedding layer. The empty feature as empty tensor with *nil* weights when the training flag is *False*. To make a parallel with human logic, it would the equivalent of saying: we know about the existence X node, but we do not know anything about it.

---

## D  Training flags for the best models

Table 6: Training flags for the best models

(a) **Objectification detection**

| Flag | Value |
|------|-------|
| Attribute | True |
| Place | True |
| Character | False |
| Context | True |
| Relationship | True |
| Emotion | False |
| Speech | True |

(b) **Interaction classification**

| Flag | Value |
|------|-------|
| Attribute | False |
| Place | False |
| Character | False |
| Context | True |
| Relationship | True |
| Emotion | True |
| Speech | True |

# E    Proposed framework for the neuro-symbolic approach

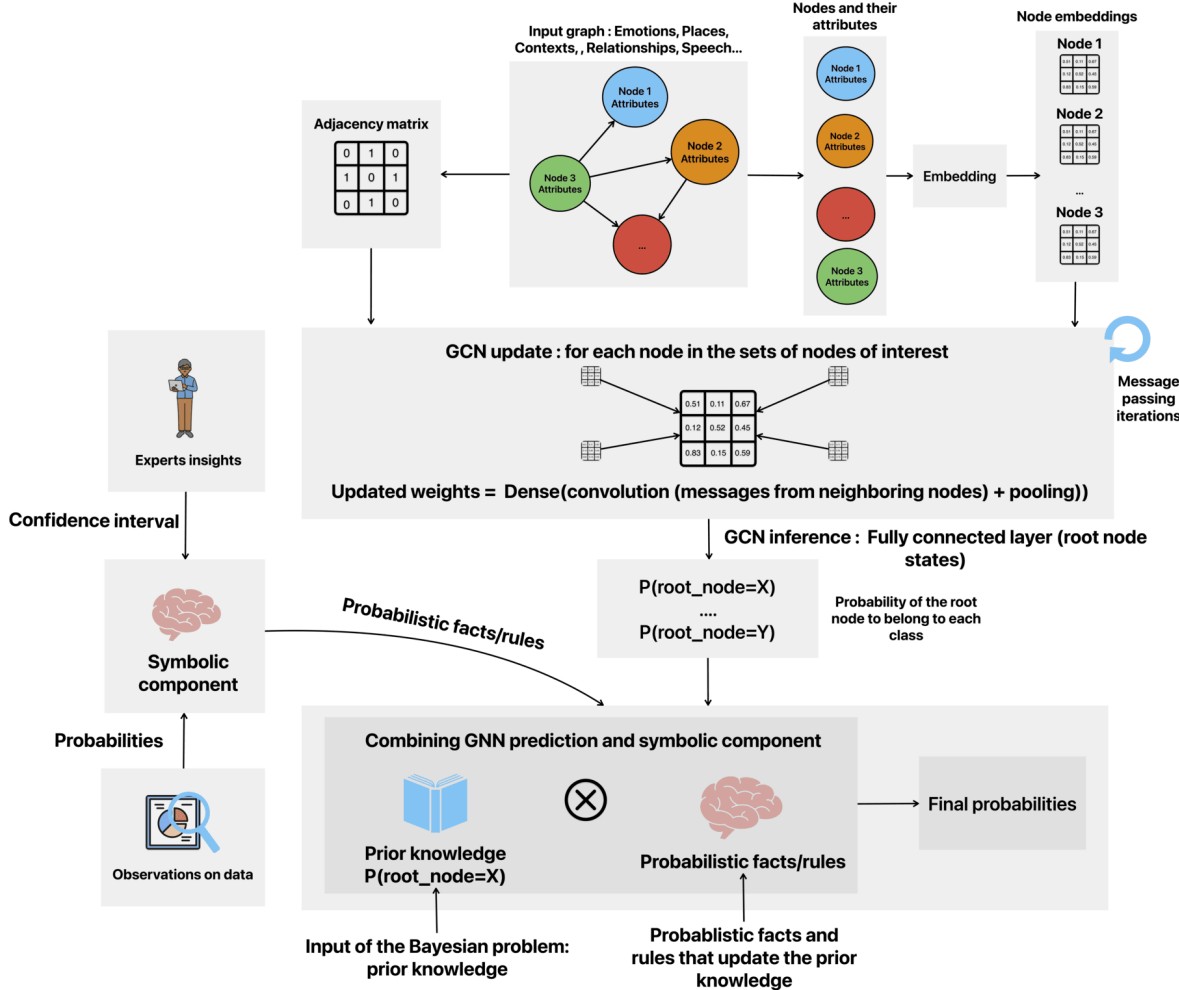

Figure 3: Integrating domain knowledge: prototype

