# OpenReview forum: "Towards context and domain-aware algorithms for scene analysis"
_TMLR — Accepted by TMLR_

### Review · Reviewer_dXtd · 2024-05-31

**Summary Of Contributions:**

This research paper proposes a framework for analyzing scenes in video content by integrating contextual data, such as character relationships and scene context, into the analysis.
They proposed adopt graph neural networks (GNNs) to process the relationships and interconnected knowledge within a scene into a three-step process: (i) Feature mapping:  Each node and edge in the graph is assigned a representation based on its type (character, scene, relationship, etc.); (ii) Graph update: The model learns the importance of different nodes and connections by iteratively updating the graph representations using different GNN architectures; (iii) Classification: Based on the learned representations, the model classifies the overall scene or individual interactions within the scene.

The framework is evaluated on two tasks: objectification detection (Tores et al 2024) and interaction classification (Vicol 2018), and claim to achieve promising results compared to baseline methods.

**Audience:**

No

**Broader Impact Concerns:**

Authors express Concerns on diverse and representative datasets, regular audits and evaluations, education and awareness and transparency and accountability. As deals with human content and interaction I suggest adding privacy protection such as to  safeguard individuals' identities and ensure ethical data handling practices throughout the research process.

**Claims And Evidence:**

No

**Requested Changes:**

Suggestions for improving the paper:

- Definition of Objectification Detection Task: Provide a clearer definition of the "objectification detection" task in the end of Section 3. The term "objectification" is often associated with negative connotations, and the task itself may not be widely known.
Dataset Description: Enhance the description of the dataset from Tores. Specify the number of objectification tags and provide some examples to aid in understanding the task.

- Method Description: For easier comprehension, start with an abstract overview of the full method. Isolate this description from hyperparameter and implementation details. Currently, Section 5 partially serves this purpose, but some components are described in Section 4. Reorganize the paper by moving Section 5 upwards and adding missing elements. Then, introduce an experimental section for implementation details.

- Figure 2 Placement: Move Figure 2 to the beginning of the method section. Include a brief description of its components (Feature map, Graph Update, etc.) to help readers understand the overall composition before delving into details.

- Clarification of the claims: are the authors presenting a novel methodology or investigating various Graph Neural Network (GNN) architectures (GCN, MPNN, Sage) for the given tasks? Table 2 seems to support the latter, while Table 4 supports the former.

- External results comparison: The comparison of the objectification detection results with external sources appears to be missing. Table 1 only includes variant comparisons. Additionally, the argument presented in Table 3 is challenging to comprehend without prior knowledge of Tores 2024. How can expert annotations be utilized to interpret the presented results? Speech is not mentioned in other tables, and the scene is mentioned in Table 3a but not in Table 3b. What is the performance of Tores et al in this task?

- Clear distinction of Tores and Vicol baseline results on the tables: in its current version is hard for the reader not familiar with these two references, where their corresponding baseline results appear on the presented tables.

**Strengths And Weaknesses:**

As a strength, the paper paper underscores the importance of contextual data for scene analysis, providing a promising framework for building more human-like AI systems capable of understanding the subtleties of social interactions.

They validate their method on two tasks. In the first, objectification detection aiming to identify scenes containing objectification, they show promising results  when combining scene graph (SG) and speech (S) and highlight the importance of contextual data for this task, as this baseline surpass these presented with SG data only.

The second task. Interaction classification, aimes to categorize interpersonal interactions, and they demonstrate that incorporating contextual information given by the SG (and also combined with dialogs) leads to better performance compared to using visual, dialogue data and face tracks in isolation or combined.

Weaknesses:
The validity of the proposal with no comparison with existing approaches is limited. The weakest aspect of the paper is comparison with existing methods and references to existent literature in the topic. The results of alternative approaches under the same tasks are minimal and/or missing. A quick search on results for the MovieGraphs dataset resulted in the following leaderboard:
https://paperswithcode.com/dataset/moviegraphs

---

### Review · Reviewer_4U97 · 2024-06-29

**Summary Of Contributions:**

This paper focuses on multimodal video classification tasks and proposes incorporating "contextual information" such as scene context, along with integrating "domain knowledge" including feature importance and simple Bayesian rules into the classification process. To achieve this, the authors utilize existing graph neural networks, integrating the contextual information as one of the graph nodes. The feature importance is emphasized by constraining the weight magnitude associated with less significant nodes, and the Bayesian rules are applied as a post-processing step following the classification by the graph neural network. The effectiveness of the proposed method is demonstrated through experiments on two video classification tasks: objectification detection and the more general interaction classification. In both tasks, the proposed method outperforms existing baselines, showcasing its effectiveness.

**Audience:**

Yes

**Broader Impact Concerns:**

The authors have discussed the Broader Impact sufficiently.

**Claims And Evidence:**

No

**Requested Changes:**

Please refer to the weakness for the required changes.

**Strengths And Weaknesses:**

Strengths:
- This paper provides a comprehensive and detailed description of the method, from data preprocessing to hyperparameter selection, which ensures reproducibility along with the supplementary code provided.

- Although I am not familiar with the tasks this paper addresses, particularly objectification detection, the authors have conducted a thorough review of previous literature to give the audience more context.

- The experiments not only achieve better performance than the selected baselines but also prove the major claims regarding the integration of contextual information and domain knowledge that were made at the beginning of the paper.

Weakness:
- First, the authors emphasize the importance of contextual information, yet they do not clearly define or provide references about what are contextual information. In the introduction, the statement "can only be achieved by incorporating contextual cues such as the context of the scene" offers very limited information. Specifically, the issue is twofold: firstly, besides the context of the scene, what other entities contain necessary context? Secondly, what specifically are these contexts, such as for a scene?
In Section 2, the authors provide examples like "relationships between the characters on-screen, or the potential speech during the interactions." In Section 8.3, further examples are given, such as visual place recognition, age and gender, as well as emotion and relationship. These examples do not clearly define contextual information, suggesting that any information could be considered as contextual information. This lack of clarity undermines the contribution of integrating contextual information since it cannot be distinctly identified. Moreover, the examples are inconsistent with the pipeline diagram in Figure 1, where the context node is connected to the scene, while character relations are linked to two characters, clearly indicating different concepts.
In summary, the authors should provide a clear definition of contextual information—what it includes and what it does not. Without this, the first contribution of the paper does not hold.

- Second, assuming there is a clear definition of contextual information, how it is integrated into the classification should also be further justified. The authors add the contextual information as a node and use existing graph neural networks. Since there is no contribution to the model itself, the authors should explain why graph neural networks are the best choice, rather than other structures such as transformers, or even off-the-shelf vision-language models (https://multimodal-vid.github.io/).

- Finally, the experiments should be improved to better support the contributions. Firstly, the discussion of different choices of graph neural networks, which could be added to the appendix, is unnecessary. Secondly, after having a clear definition of contextual information, the authors should not only provide sensitivity scores for each type of contextual feature but also conduct ablation studies excluding each type of contextual information. Lastly, the claim of limited contribution from image features is unconvincing. Instead, visual input is important as some of the contextual information and the information for the naive Bayesian post-processing are derived from visual features. Therefore, rather than simply using existing feature extractors in parallel, I would suggest the authors extract different types of information from visual features using the same backbone feature maps.

---

### Review · Reviewer_3bJ8 · 2024-10-22

**Summary Of Contributions:**

This paper presents an approach to scene analysis that incorporates contextual data into graph neural networks. The authors evaluate their methodology on two tasks: objectification detection and interaction classification in video content. The work introduces methods for integrating domain expertise and provides interpretability insights through sensitivity analysis of the model's decision-making process.

**Audience:**

Yes

**Broader Impact Concerns:**

The authors have list broader impact in the Broader Impact Statement Section.

**Claims And Evidence:**

Yes

**Requested Changes:**

My major concern with this paper is the insufficient detail provided in the methodology description. This lack of detail makes it difficult to understand the model architecture and reproduce the results. The authors need to provide more comprehensive information in the manuscript to ensure readers can fully understand and implement the approach.

**Strengths And Weaknesses:**

Strengths:

The paper proposes a novel method for scene analysis by explicitly incorporating contextual information.

The experimental evaluation is comprehensive, with ablation studies and detailed analysis.

Experimental results show improvements over baseline approaches.

Weaknesses:

It is unclear how scene elements are mapped to interaction graphs. I was unable to understand how characters are identified in videos, what attributes are tracked, and how they're extracted. It is also unclear how scene and context information are extracted. The embedding process is also unclear, as there are no details on whether or how different node types are embedded differently.

 The details provided about the datasets are insufficient. For the ObjectifyGaze Dataset, the authors need to briefly introduce the source of the videos and the lengths of the scenes. For the MovieGraphs Dataset, the authors need to introduce the characteristics of the 51 movies, the lengths of the scenes, and the interaction classes involved.

---

> ### Author Response · Authors · 2024-10-29
> **Addressing the concerns of reviewer 3bJ8**
>
> Dear reviewer,
>
> Thank you for your valuable feedback and for taking the time to review our paper. We have carefully considered your comments and have made revisions to address the concerns you raised. The changes include:
>
> - **More details on the original datasets, including the elements within the graphs**. However, since scenes contain varying lengths that can range from a few seconds to a whole minute, we could not devise a generic way to present this information consistently across both datasets.
> - **More details on the methodology in Figure 2**, allowing for better visualization of the shapes and the embedding process.
>
> Given that there are more than 101 interaction classes, we provide some samples to the reader and suggest that they refer to the original dataset for more information on the exact classes for better readability.
>
> In response to your questions:
> - Characters within the scenes are represented as Character nodes; we do not identify them individually.
> - No attributes are tracked or extracted, as they are provided by the original dataset. We have added few examples of attributes in the new version.
> - As stated in Section 5.1 of the new version, nodes are embedded using specific flags denoting the absence (when false) or presence (when true) of all their attributes during the embedding process. The attributes of a specific node are concatenated and then passed through an embedding layer, as shown in the new version of the figure.
>
> We hope that these revisions have addressed your concerns and we look forward to your continued feedback.

---

### Review · Reviewer_TMWG · 2024-12-13

**Summary Of Contributions:**

The paper studies a graph-centric deep learning approach for scene analysis. For this, the authors construct an interaction graph containing the context and interactions. Based on this graph, the authors devise a deep learning architecture with graph neural networks at its core to tackle two tasks: (a) objectification detection and (b) link classification. The core aim of this work is to provide a human-like understanding of interactions between humans. It is one of the few works in the field that attempts to incorporate contextual data for the interpretation of scenes.

**Audience:**

Yes

**Broader Impact Concerns:**

No concerns

**Claims And Evidence:**

Yes

**Requested Changes:**

1. The paper could be more explicit about the targets in the tasks. I assume objectification detection is a graph classification task while interaction classification is an edge-level classification task. Perhaps one example per task would also be helpful (e.g., added to the appendix).
2. The empirical evaluation could be more comprehensive in comparing to other approaches or the authors should elaborate on why there are not alternative approaches.
3. The related work section only contains 9 references. Is this really comprehensive? If so, the authors should be explicit about why there are only so few relevant works.

## Minor
- Why past tense in 5.4 "we introduce**d** an interpretability"
- It is confusing that Figure 2 follows page 6, given that the reader expects an equation next. Perhaps move Eq 4 to the top of page 7, there is plenty of space
- It is uncommon to use * for multiplication. Consider \cdot instead. Also, in the equations, the authors should escape the math mode for terms like "TPR," "Precision," etc.

**Strengths And Weaknesses:**

## Strengths
1. Intuitive and well-motivated approach for utilizing graph machine learning for the task of scene understanding
2. The paper is easy to follow and contains a good amount of technical details that is often missing in other papers, providing a good overview over the method and data preprocessing etc.

## Weaknesses
1. The lack of visual features is somewhat concerning to claim human-like understanding
2. The empirical evaluation compares only with a few (old) baselines
3. The paper introduces some interpretability features but lacks their empirical evaluation

---

> ### Author Response · Authors · 2024-12-13
> **Adressing the concerns of Reviewer TMWG**
>
> Dear reviewer,
>
> Thank you for your valuable feedback and for taking the time to review our paper. In response to your concerns:
>
> - We have added a specification to the task description paragraphs (Section 3.1 and Section 3.2) : both are node-level classification tasks.
> - The benchmark for detecting objectification using these scenes is very recent, initiated this year (2024). Therefore, we could only compare our work with the baseline study from Tores et al. Moreover, there is limited exploration of both context integration in scene analysis and GNNs for scene analysis in the literature, particularly studies closely related to our work. Additionally, few studies have addressed the interaction classification task from MovieGraphs, with the best results achieved by the baseline study we selected, LIREC. While some related works were cited in the introduction to provide the global context of our research, it was essential to do so for depicting the whole picture.
> - To assess our sensitivity scores calculation, we initially compare the sensitivity computed on the entire dataset with the top 4 reasons—why a scene is considered objectifying or not—as indicated by expert annotators of the original dataset, in Table 3.a and 3.b. Following this, in the small ablation study after section 6.3.2, we present experiments with specific cues (nodes) removed. The results align closely (though not perfectly) with our calculations and comparative analysis.
>
> We have appended a new version with major changes highlighted in green, and minor concerns such as multiplications and past tense corrections have been addressed. We hope that these revisions have addressed your concerns and we look forward to your continued feedback.

---

### Comment · Reviewer_1PiH · 2024-05-03
**Not my expertise.**

Not my area of expertise, please un-assign!

---

> ### Comment · Action_Editor_vzeJ · 2024-05-15
>
> I removed you from the reviewer list.

---

### Comment · Reviewer_dnnU · 2024-07-01
**Not my expertise**

Dear meta-reviewer,

I have the same comment as the other reviewer. This submission is very far from my area of expertise. Therefore, I can not provide a qualified review for it. Please remove me from the reviewer list for this paper.

---

> ### Comment · Action_Editor_vzeJ · 2024-07-02
> **I removed you from the list of reviewers.**
>
> I removed you from the list of reviewers.

---

### Decision · Action_Editor_vzeJ · 2025-01-07

**Recommendation:** Accept with minor revision

**Comment:**

This paper presents a graph learning approach for scene analysis. The method integrates contextual information using GNNs for two tasks: Objectification detection and Interaction classification.

The paper was reviewed by expert reviewers and the summary of their findings are as follows:

Strengths
- Incorporating contextual cues and graph-based modeling is a promising direction for scene understanding.
- The paper provides detailed technical descriptions that facilitate reproducibility.
- There are extensive ablation studies demonstrating the significance of contextual data for the tasks.

Weaknesses
- The lack of a clear and consistent definition of "contextual information" undermines the core claim of the paper.
- Comparisons with existing state-of-the-art methods on the datasets are partially insufficient.
- The dataset descriptions, particularly ObjectifyGaze, lack sufficient detail for readers to assess the validity of the evaluation.

Requested changes in the revision:
- Authors should provide a clear and consistent definition of "contextual information" and its role in the model. Clearly outline how scene elements are mapped to interaction graphs, including character identification and node embeddings. Add a high-level overview of the method before delving into implementation details.
- Authors should enhance Dataset Descriptions. For ObjectifyGaze, describe the source of videos, the objectification tags, and examples. For MovieGraphs, provide details on the dataset’s composition and interaction classes.
- Authors should address all stated minor issues

This paper proposes an interesting method for scene understanding but requires revisions to ensure clarity and rigor. Once these changes are made, the paper will make a valuable contribution to the field.

Note: some of these changes are already made in the revision during the discussion period and kept here for the completeness of the summary.

**Audience:**

There is a clear audience within the fields of graph neural networks and scene understanding communities which would find this paper interesting.

**Claims And Evidence:**

The claims made in the submission are supported by evidence, but there are gaps in clarity and comprehensiveness which can be fixed during the revision.

The paper demonstrates improvements in the tasks of objectification detection and interaction classification. Its use of graph neural networks (GNNs) to model contextual information is well-motivated and supported by the empirical evidence.

The exposition can clearly be improved very significantly as some details are not clear. Most of these are handled during the discussion phase but a new revision is needed.